# Guanine Radicals Generated in Telomeric G-Quadruplexes by Direct Absorption of Low-Energy UV Photons: Effect of Potassium Ions

**DOI:** 10.3390/molecules25092094

**Published:** 2020-04-30

**Authors:** Evangelos Balanikas, Akos Banyasz, Gérard Baldacchino, Dimitra Markovitsi

**Affiliations:** 1LIDYL, CEA, CNRS, Université Paris-Saclay, F-91191 Gif-sur-Yvette, France; vangelis.balanikas@cea.fr (E.B.); akos.banyasz@ens-lyon.fr (A.B.); gerard.baldacchino@cea.fr (G.B.); 2Univ Lyon, ENS de Lyon, CNRS UMR 5182, Université Claude Bernard Lyon 1, Laboratoire de Chimie, F-69342 Lyon, France

**Keywords:** guanine-quadruplexes, DNA, photo-ionization, oxidative damage, electron holes, guanine radicals, reaction dynamics, time-resolved spectroscopy

## Abstract

The study deals with the primary species, ejected electrons, and guanine radicals, leading to oxidative damage, that is generated in four-stranded DNA structures (guanine quadruplexes) following photo-ionization by low-energy UV radiation. Performed by nanosecond transient absorption spectroscopy with 266 nm excitation, it focusses on quadruplexes formed by folding of GGG(TTAGGG)_3_ single strands in the presence of K^+^ ions, **TEL21/K^+^**. The quantum yield for one-photon ionization (9.4 × 10^−3^) was found to be twice as high as that reported previously for **TEL21/Na^+^**. The overall population of guanine radicals decayed faster, their half times being, respectively, 1.4 and 6.7 ms. Deprotonation of radical cations extended over four orders of magnitude of time; the faster step, concerning 40% of their population, was completed within 500 ns. A reaction intermediate, issued from radicals, whose absorption spectrum peaked around 390 nm, was detected.

## 1. Introduction

**G**-quadruplexes are four-stranded structures formed by guanine (**G**) rich DNA/RNA strands in the presence of cations such as K^+^ and Na^+^, encountered in cells. They are characterized by the vertical stacking of **G** tetrads in which **G**s are interconnected via Hoogsteen hydrogen bonds (Figure 1). **G**-quadruplexes play a key role in important biological functions [1] and constitute therapeutic targets [2,3]. In addition, they are promising for applications in the field of nanotechnology [4], exploiting, for example, their ‘molecular wire’ behavior [5], based on the transport of electron holes [6,7]. In view of their biological importance and potential technological impact, characterizing the generation and fate of guanine radical cations (**G**)^+●^ (electron holes) in **G**-quadruplexes is essential.

A large number of articles reports oxidative damage and/or charge transport in **G**-quadruplexes, in which **G** radical generation is mediated by other molecules. In contrast to these studies, our group discovered recently an unexpected path leading to the radical formation in **G**-quadruplexes: photo-ionization by low-energy UV photons [8]. The term ‘low-energy’ is used in comparison to the ionization potential of the various components of nucleic acids, which preclude vertical electron photo-detachment upon irradiation around the absorption maximum (~260 nm). Yet, one-photon ionization quantum yields (φ_1_) ranging from 3.5 × 10^−3^ to 8.1 × 10^−3^ were determined at 266 nm for four different **G**-quadruplex structures [8,9,10,11,12]. These structures were formed by folding of the telomeric sequences GGG(TTAGGG)_3_ and TAGGG(TTAGGG)_3_TT in the presence of Na^+^ ions (**TEL21/Na^+^** and **TEL25/Na^+^**, respectively) or association of four single strands TGGGGT in the presence of Na^+^ or K^+^ ions ((**TG_4_T**)_4_/**Na^+^** and **(TG_4_T)_4_/K^+^**, respectively). The φ_1_ values, gathered in Table 1, indicate small variations with the number of strands composing the structure or the ending groups, but quite significant increase when Na^+^ ions are replaced by K^+^. This cation effect was observed for tetramolecular quadruplexes formed by the same sequence and exhibiting the same topology [9,11,12]. Therefore, it was important to explore whether a similar φ_1_ enhancement also occurs for other types of **G**-quadruplexes. This was the first objective of the present study.

The above-mentioned studies were performed by transient absorption spectroscopy using not only low-energy (266 nm) but also low-intensity excitation (5 ns laser pulses, incident intensity lower than 2 × 10^6^ Wcm^−2^). These conditions, associated with the experimental protocols described in Section 3, allowed us to explore the spectral and dynamical features of guanine radicals in the absence of additives other than the phosphate buffer in which **G**-quadruplexes are dissolved. We showed that two different processes take place. First, deprotonation of radical cations occurs in the position 2, giving rise to (**G**-H2)^●^ radicals instead of the (**G**-H1)^●^ radicals (Figure 1a), which are observed for the dGMP mononucleotide [13] and various duplexes [10,14,15]. Such a different deprotonation route, which was initially detected by photosensitized oxidation of **G**-quadruplexes [16], is due to the fact that the proton at position 1 is blocked by a Hoogsteen hydrogen bond (Figure 1a). Another process, (**G**-H2)^●^ → (**G**-H1)^●^ tautomerization, was observed on the millisecond time scale for both **TEL21/Na^+^** [8] and **(TG_4_T)_4_/Na^+^ [9]**. In contrast, we found that tautomerization is completely inhibited in the case of **(TG_4_T)_4_/K^+^**, showing that K^+^ ions modify the reactivity of guanine radicals [11,12]. This is an important issue, because, in principle, each type of radical should lead to different final lesions, although those resulting from (**G**-H2)^●^ have not been identified so far. In view of these considerations, the second objective of the present study was to explore whether the suppression of (**G**-H2)^●^ → (**G**-H1)^●^ tautomerization in the presence of K^+^ ions is a general rule.

Here, we focus on **G**-quadruplexes formed by folding of the telomeric sequence GGG(TTAGGG)_3_ in the presence of K^+^ ions, **TEL21/K^+^**. The results were compared with those obtained for **TEL21/Na^+^**, published in 2017 [8]. However, unlike the tetramolecular **G**-quadruplexes studied previously, whose structure does not change when Na^+^ ions are replaced by K^+^ ions, those formed by the human telomeric sequences undergo drastic geometrical rearrangements. NMR experiments on **G**-quadruplexes formed by sequences containing four telomer repeats TTAGGG in the presence of K^+^ ions evidenced the coexistence of two hybrid forms in equilibrium [17,18]. The two forms differed in the successive order of the loop arrangement and the strand orientation. The favored hybrid form depends on the capping groups. As the objective of our work was to examine whether the effect of K^+^ ions on the photoionization, on the one hand, and the reactivity of deprotonated radicals, on the other, found for tetramolecular **G**-quadruplexes is also encountered for **TEL21/K^+^**, the exact structure adopted by the latter system is not an issue.

## 2. Results and Discussion

### 2.1. Photo-Ionization

Photo-ionization was studied by monitoring the hydrated ejected electrons (e_hyd_^−^), whose absorption spectrum is well-known [19]. These species disappear within 2 µs through a reaction with the phosphate buffer. Their initial concentration [e_hyd_^−^]_0_ was determined as the ratio A_1_/ε, where A_1_ was derived from the fit of the decay at 700 nm with a mono-exponential function (Figure 2a) and ε represented the molar absorption coefficient (19,700 mol^−1^Lcm^−1^) [19].

The ionization curve shown in Figure 2b was obtained by varying the intensity of the incident laser pulses and determining the [e_hyd_^−^]_0_. The experimental points were fitted with a linear model function: [e_hyd_^−^]_0_/[hν] = φ_1_ + α [hν]. The intercept (φ_1_) on the vertical axis provided the quantum yield for one-photon ionization while the slope α was associated with two-photon ionization. The φ_1_ derived for **TEL21**/**K^+^** from Figure 2b, (9.4 ± 0.1) × 10^−3^, was twice as high compared to that reported for **TEL21**/**Na^+^**, (4.5 ± 0.6) × 10^−3^ [8]. Thus, replacing Na^+^ by K^+^ ions enhanced by a factor 2 the propensity of telomeric **G**-quadruplexes to photo-eject an electron, as in the case of tetramolecular systems (Table 1).

The mechanism proposed to explain the φ_1_ enhancement is related to a non-vertical electron photo-detachment and is described in reference [11]. Briefly, charge transfer (CT) states are formed during the excited state relaxation [20,21,22]. A small fraction of the CT states may undergo charge migration, followed by charge separation. Subsequently, the electron is ejected from the negatively charged base. In the frame of this scenario, K^+^ ions, which are larger and less mobile than Na^+^, should hinder the stabilization of CT states, favoring charge separation instead. This is corroborated by a recent time-resolved study focusing on the excited states, which shows that the transient absorption of CT states is more important for **TEL21/Na^+^** compared to **TEL21/K^+^** [22].

### 2.2. Transient Species

The transient absorption spectra recorded for **TEL21/K^+^** on the microsecond and millisecond time-scales, after the decay of hydrated electrons, exhibit significant evolution. This is illustrated in Figure 3, where the spectra obtained at selected times are presented after normalization of their intensity at 500 nm so that to better perceive the changes in their profile. The most striking modifications are (i) an intensity decrease around 600 nm and (ii) an increase at 350–450 nm. The former evolution was already observed for **TEL21/Na^+^** and attributed to the disappearance of (**G**-H2)^●^ radicals [8], characterized by an absorption band peaking around 600 nm [23,24]. In contrast, no substantial increase at 350–450 nm was detected in the **TEL21/Na^+^** transient spectra.

In order to get an insight into the species responsible for the transient absorption in Figure 3, we considered the spectra of the three types of radicals reported in the literature for monomeric guanosines [23,24]. Our previous studies showed good agreement between the monomeric radical spectra and those determined for **G**-quadruplexes, in particular in the visible spectral domain [8,9]. Thus, we use linear combinations of the monomer radical spectra to reconstruct the time-resolved spectra of **TEL21/K^+^**. In addition, we got information about the dynamics of radical deprotonation and tautomerization processes by observing transient signals at wavelengths better reflecting each process. Indeed, inspection of the spectral differences corresponding to pairs of monomer radicals (Figure 4) led to several interesting observations: (i) for all radical transformations, the differential molar absorption coefficient ∆ε had a minimum value (|∆ε| < 100 mol^−1^Lcm^−1^) at 510–515 nm, i.e., the ε values are quite similar (≈1500 mol^−1^Lcm^−1^) for all three radicals. Consequently, the ∆A at this wavelength provides a good estimate of the total radical population. (ii) (**G**)^+●^ → (**G**-H2)^●^ deprotonation can be better followed around 400 and 620 nm. (iii) The spectra of (**G**)^+●^ and (**G**-H1)^●^ are quite similar in the visible domain but can be differentiated in the near infra-red. (iv) (**G**-H2)^●^ → (**G**-H1)^●^ tautomerization can be detected at 720 nm. Changes in the UV spectral range are not informative because, on the one hand, **G**-quadruplexes absorb at longer wavelengths compared to monomeric guanosine [8], and, on the other, photon absorption may induce formation of photo-dimers [8,21]; some of them absorb in the UVA region, and their dynamics intervenes on the millisecond time-scale [25,26].

We found that the total radical population at 3 µs [R]_3µs_, determined from the ∆A at 512 nm and using an ε value of 1500 mol^−1^Lcm^−1^, equals the initial concentration of hydrated electrons. Therefore, we concluded that, within the precision of our measurements (±5%), all the generated radicals were still present at 3 µs, that is [R]_3µs_ = [R]_0_. Accordingly, the transient absorption signals at 512 nm shown in Figure 5, where the intensity at 3 µs is normalized to 1, represents the survival probability P_t_ of the total radical population at time t, i.e., the fraction of the initial radical concentration at time t: P_t_ = [R]_t_/[R]_0_. For a quantitative description, the experimental traces recorded on the microsecond and millisecond time-scale were fitted with bi-exponential functions, although such fits have no physical meaning [10]. Thus, we found that the radical decay spans more than four orders of magnitude; at 50 µs, 80% of the radical population is present, while at 10 ms, this percentage dwindles down to 20%. The time needed for the radical population to be reduced by a factor of two (τ_1/2_) was 1.4 ms. A much longer τ_1/2_ value is determined for **TEL21/Na^+^** (6.6 ms; Appendix A).

In Figure 6 we present the transient absorption spectra determined for **TEL21/K^+^** at 3 µs, 50 µs, 1.4 ms, and 10 ms, expressing their intensity as ∆A/[R]_t_, where [R]_t_ = P_t_ [R]_0_. The units correspond to molar absorption coefficient, allowing quantitative comparison with the spectra of monomeric radicals. Thus, we reconstructed each transient spectrum by a linear combination of the monomer radical spectra using a single variable, as explained below.

The transient absorption spectrum of **TEL21/K**^+^ at 3 µs (P_t_ = 1) is well described by a linear combination of the monomeric (**G**)^+●^ and (**G**-H2)^●^ radicals at a ratio 0.6/0.4. A similar ratio (0.5/0.5) was found for **TEL21/Na^+^** [8]. Figure 7 shows the transient signals obtained at 400 and 620 nm for N_2_O saturated solutions of **TEL21/K^+^**. This latter compound quenched very efficiently the hydrated electrons, allowing the observation of transient absorption signals associated solely with radicals. The dynamics of the deprotonation process taking place before 3 µs and concerning 40% of the radical population is illustrated in Figure 7. It is reflected in the transient signals recorded at 400 and 620 nm. As predicted from the difference spectrum corresponding to (**G**)^+●^ → (**G**-H2)^●^ deprotonation (Figure 4), we observe a decay at 400 nm and a rise at 620 nm (Figure 7). No rise was observed at 720 nm, justifying that (**G**-H1)^●^ radicals were not taken into account in the reconstruction of the 3 µs spectrum (see Table 2). Both signals in Figure 7 can be described by mono-exponential functions with the same time constant of 0.14 µs. We correlate this fast deprotonation process with radical cations located in tetrads in **TEL21/K**^+^ that are most exposed to bulk water.

Before reconstructing the spectrum at 50 µs, we observe that the transient signal at 720 nm exhibited a rise completed within 20 µs (Figure 8). This rise is associated with the formation of the (**G**-H1)^●^ radical. Its concentration, estimated using the ∆ε from Figure 4 (550 mol^−1^Lcm^−1^), corresponding to 5% of [R]_0_. This knowledge allows us to reconstruct the transient spectrum using as a unique variable, the concentration ratio [(**G**)^+●^]/[(**G**-H2)^●^]. Thus, the total radical concentration [R]_50 µs_ at 50 µs (P_t_ = 0.80), can be decomposed, in terms of molar fractions of [R]_0_, as (**G**)^+●^ (0.40), (**G**-H2)^●^ (0.35) and (**G**-H1)^●^ (0.05). The transient spectra at 1.4 ms and 10 ms can be simply described by the (**G**-H1)^●^ spectrum with molar fractions of 0.50 and 0.45, respectively (see Table 2). Despite the errors inherent in this type of analysis, it clearly appears that (**G**-H2)^●^ → (**G**-H1)^●^ tautomerization does take place. Therefore, we conclude that the simple presence of K^+^ ions is insufficient to suppress this reaction, which probably depends on other dynamical and structural factors.

The transient absorption spectrum at 30 ms, when the major part of radicals has disappeared, exhibits a peak at 320 nm and a shoulder at the 360–390 nm area (Figure 9a). The latter spectral features are correlated with reaction intermediates and/or final reaction products. The spectral signature of final reaction products can be identified in the steady-state differential absorption spectrum, determined by subtracting the spectrum recorded prior irradiation from that recorded after irradiation. This differential spectrum is also shown in Figure 9, where ∆A was normalized by the concentration of photons absorbed in the probed volume. The two spectra are clearly different, not only in their profile but also in their intensity, the one at 30 ms being seven times more intense. The transient signal at 365 nm (Figure 9b) shows a slow decay, lasting for more than 180 ms, which is the longest time that we can determine. Accordingly, we conclude that we were dealing with a reaction intermediate.

In our experiments, reaction intermediates may have originated either from **G** radicals or from photochemical reactions, being formed at the potential energy surface of an excited state. This is the case, for example, of an adenine dimer, determined on the millisecond time scale in the case of adenine tracts [27], in parallel with that of adenine radicals [28]. The concentration of the chemical species issued from radicals is proportional to ejected electrons, while that related to photoreactions is proportional to the number of absorbed photons. We found that the transient absorption signals at 365 nm, bearing fingerprints of both radicals and the unknown reaction intermediate, are not altered when the ratio [e_hyd_^−^]_0_/[hν] increases by 50%, from 0.017 to 0.026 (Appendix A). Not only the excitation intensity but also the presence of oxygen has no effect on the decays (Appendix A); this observation concerns both the radical decays and those of the reaction intermediate and final photoproduct. Regarding the latter, we remark that the steady-state differential absorption spectrum determined for **TEL21/K^+^** was similar to that reported for **TEL21/Na^+^** (Appendix A), which was tentatively attributed to guanine dimers [21]. However, in the case of **TEL21/K^+^**, the spectral intensity, normalized by the number of absorbed photons, was three times lower compared to that of **TEL21/Na^+^**, suggesting a lower quantum yield of the associated photo-reaction.

## 3. Materials and Methods

GGG(TTAGGG)_3_ oligomers, purified by reversed-phase HPLC and tested by MALDI-TOF, were provided by Eurogentec Europe (Liège, Belgium) as a lyophilized powder. They were dissolved in phosphate buffer (0.15 mol L^−1^ KH_2_PO_4_, 0.15 mol L^−1^ K_2_HPO_4_); the purity of both potassium salts (Fluka) used for the buffer was 99.99%. Solutions were prepared using ultrapure water delivered by a Millipore (Milli-Q Integral) system (Merck, Millipore SAS, Guyancourt, France). The pH, measured by a pH 210 apparatus (Hanna Instruments, Tanneries, France), was adjusted to 7 by the addition of a concentrated KOH solution. **G**-quadruplexes were prepared by heating 2 mL mother solution (oligomer concentration: ~10^−3^ molL^−1^) to 96 °C during 5 min in a dry bath (Eppendorf-ThermoStatplus, Merck, Darmstadt, Germany); then, the temperature was decreased to the melting point (73 °C; Appendix A) where it was maintained for 10 min; finally, the solution was cooled to 4 °C (cooling time. 2 h), where it was incubated overnight. Prior to time-resolved experiments, **TEL21/K^+^** solutions, kept in −20 °C, were heated to 40 °C, to destroy possible higher-order aggregates [29], cooled slowly to room temperature and their absorbance was adjusted 2.5 ± 0.2 over 1 cm. A typical melting curve is shown in SI-4. During the experiments, 2 mL of solution, contained in a 1 cm × 1 cm quartz cell (Hellma-France, Meaux, France), was mildly stirred while the temperature was maintained at 23 °C. To avoid re-excitation of damaged **G**-quadruplexes, solutions were replaced frequently. About 300 mL of solution, stemming from three different batches, were used for the entire study.

Steady-state absorption spectra were recorded using a Shimadzu UV-2600 spectrophotometer (Shimadzu France, Noisiel, France). The transient absorption setup used as excitation source was the fourth harmonic of a nanosecond Nd:YAG laser (Quanta Ray, Spectra Physics, Santa Clara, CA, USA). The excited area at the surface of the sample was 0.6 × 1.0 cm^2^. The analyzing beam, orthogonal to the exciting beam, was provided by a 150 W Xe-arc lamp (OSRAM XBO, Munich, Germany); its optical path length through the sample was 1 cm, while its thickness was limited to 0.1 cm in order to use the most homogeneous part of the light. The probed volume was located in the very first 0.1 cm part of the cell along the propagation of the exciting laser beam, delimited by four dedicated slits. Then, the analyzing light was dispersed in a Jobin-Yvon SPEX 270M monochromator (Horiba Jobin Yvon, Longjumeau, France), detected by a Hamamatsu R928 photomultiplier (Hamamatsu Photonics France, Massy, France) and recorded by a Lecroy Waverunner oscilloscope-6084 (Teledyne Lecroy, Courtaboeuf, France). For measurements on the sub-microsecond scale, the Xe-arc lamp (Applied Photophysics, Leatherhead, UK) was intensified via an electric discharge. Transient absorption spectra were recorded using a wavelength-by-wavelength approach. Fast shutters were placed in the path of both laser and lamp beams; thus, the excitation rate was decreased from 10 Hz to 0.2 Hz, and exposure to the analyzing beam could be minimized. The incident pulse energy at the surface of the sample was measured using a NIST traceable pyroelectric sensor (Nova2/PE25, Ophir Spiricon Europe GmbH, Darmstadt, Germany). In addition, the absorbance of the naphthalene triplet state, whose quantum yield in cyclohexane is 0.75, served as actinometer [30].

## Figures and Tables

**Figure 1 molecules-25-02094-f001:**
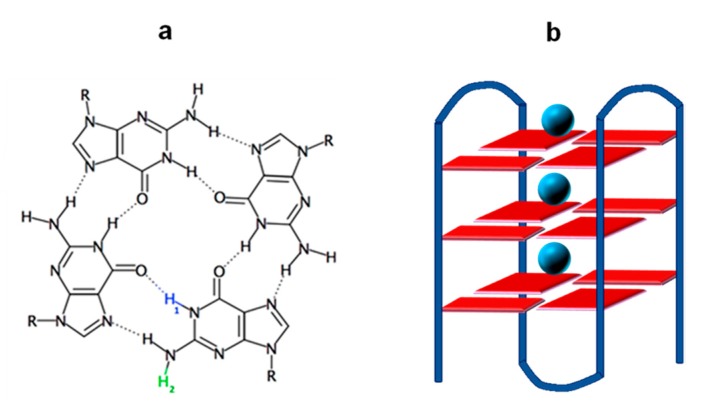
Schematic illustration of (**a**) the **G**-tetrad and (**b**) a **G**-quadruplex formed by folding of a single strand; blue circles represent metal ions located in the central cavity of the **G**-quadruplex. Transfer of the blue or green proton (**a**) to the aqueous solvent gives rise to (**G**-H1)^●^ or (**G**-H2)^●^ deprotonated radicals, respectively.

**Figure 2 molecules-25-02094-f002:**
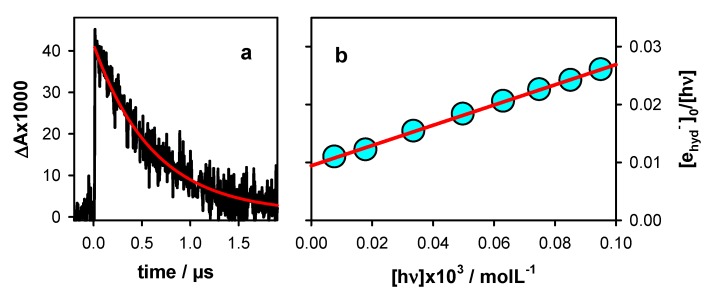
Hydrated electrons stemming from photo-ionization of **TEL21/K^+^**. (**a**) Transient absorption signal recorded at 700 nm with an incident excitation intensity of 1.7 × 10^6^ Wcm^−2^. (**b**) Ionization curve; [e_hyd_^−^]_0_ and [hν] denote, respectively, the zero-time concentration of hydrated ejected electrons and the concentration of absorbed photons per laser pulse. Red lines represent fits with (**a**) a mono-exponential A_0_ + A_1_exp(−t/τ_1_) and (**b**) a linear [e_hyd_^−^]_0_/[hν] = φ_1_ + α[hν] model function.

**Figure 3 molecules-25-02094-f003:**
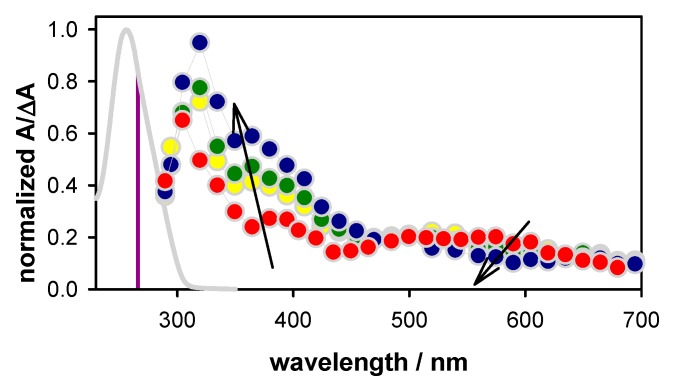
Transient absorption spectra determined for aerated solutions of **TEL21/K^+^** (circles) at 3 µs (red), 150 µs (yellow), 1 ms (green), and 6 ms (blue). ∆A at 500 nm was set to 0.2. The gray line corresponds to the steady-state absorption spectrum, and the vertical violet line indicates the excitation wavelength.

**Figure 4 molecules-25-02094-f004:**
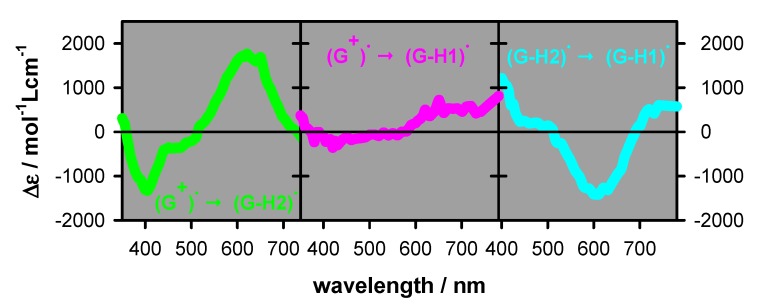
Difference spectra corresponding to (**G**)^+●^ → (**G**-H2)^●^, (**G**)^+●^ → (**G**-H1)^●^ and (**G**-H2)^●^ → (**G**-H1)^●^ processes. They were determined from the spectra of monomeric guanosine radicals [23,24]. ∆ε < 0 will be reflected by a decay in the transient absorption signal while ∆ε > 0 by a rise.

**Figure 5 molecules-25-02094-f005:**
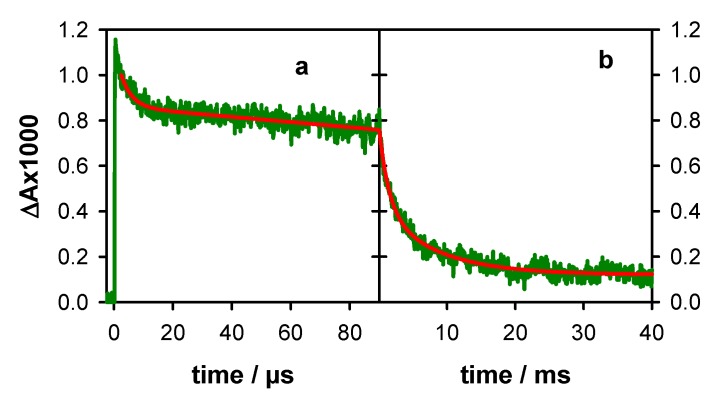
Survival probability P_t_ of the total **G** radical population in **TEL21/K^+^**, estimated by the transient absorption signals at 512 nm, recorded on the microsecond (**a**) and the millisecond (**b**) time-scales. Red lines correspond to fits with bi-exponential functions.

**Figure 6 molecules-25-02094-f006:**
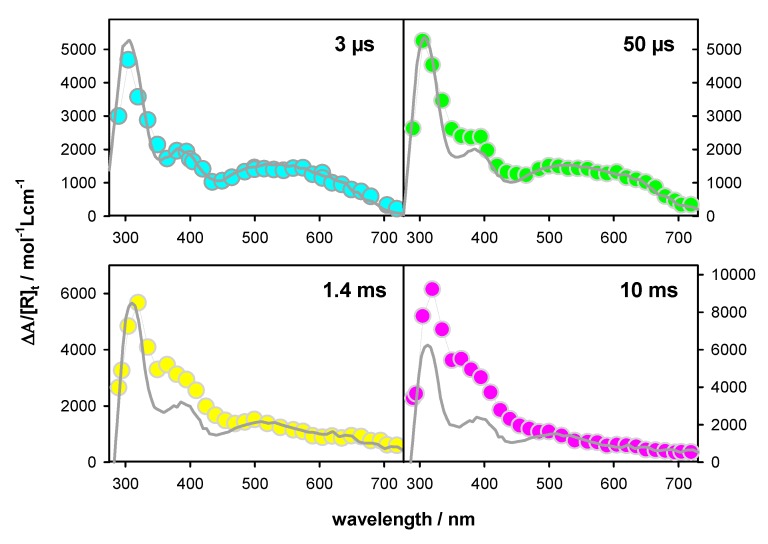
Transient absorption spectra determined for **TEL21/K^+^** at 3 µs, 50 µs, 1.4 ms, and 10 ms; ∆A was divided by the total radical concentration surviving at the considered time [R]_t_ (see Table 2). In gray: linear combinations of the monomer radical spectra considered with their ε values.

**Figure 7 molecules-25-02094-f007:**
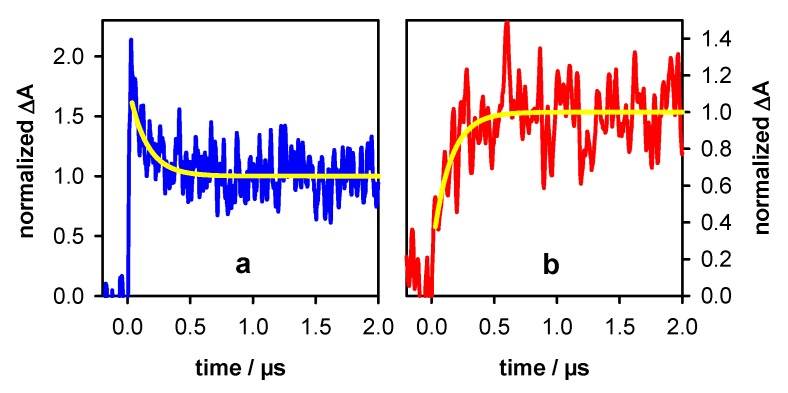
Transient absorption signals recorded at 400 nm (**a**) and 620 nm (**b**) for N_2_O saturated solutions of **TEL21/K^+^**. Yellow lines correspond to mono-exponential decay and rise with the same time constant of 0.14 µs. ∆A was normalized to 1 at 2 µs.

**Figure 8 molecules-25-02094-f008:**
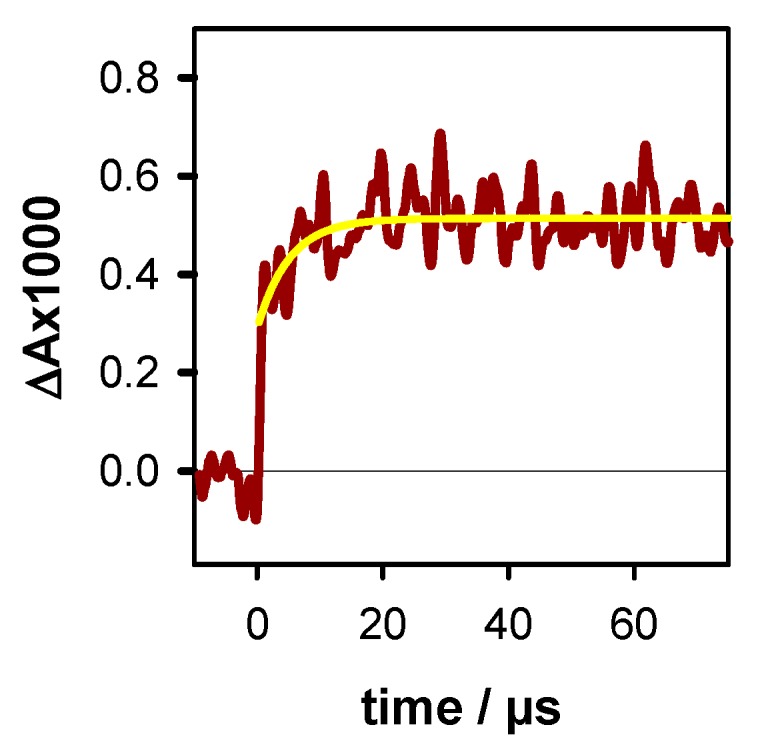
Formation of the (**G**-H1)^●^ radical in **TEL21/K^+^**, estimated by the transient absorption signal at 720 nm. The yellow line is the fit with a mono-exponential model function. N_2_O saturated solutions.

**Figure 9 molecules-25-02094-f009:**
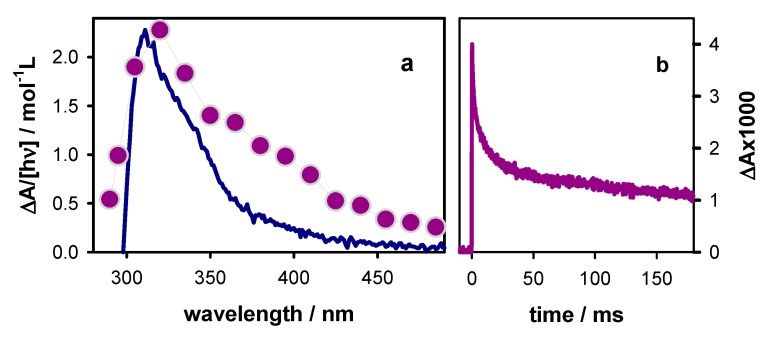
(**a**) Steady-state (blue line) and time-resolved (violet circles; 30 ms; intensity divided by 7) differential absorption spectra determined for **TEL21/K^+^**; the steady-state spectrum was obtained by subtracting the absorption spectrum recorded after irradiation with 266 nm pulses from that recorded before irradiation. [hν] is the total concentration of absorbed photons in the probed volume. (**b**) Transient absorption trace at 365 nm recorded with incident excitation intensity of 2 × 10^6^ Wcm^−2^.

**Table 1 molecules-25-02094-t001:** Quantum yields (φ_1_ × 10^3^) determined at 266 nm for one-photon ionization of **G**-quadruplexes.

TEL21/Na^+^ [8]	TEL25/Na^+^ [10]	(TG_4_T)_4_/Na^+^ [9]	(TG_4_T)_4_/K^+^ [11,12]	TEL21/K^+^
4.5 ± 0.6	5.2 ± 0.3	3.5 ± 0.5 [9]	8.5 ± 0.5	9.4 ± 0.1

**Table 2 molecules-25-02094-t002:** Evolution of guanine radical populations in **TEL21/K^+^**, expressed as molar fractions of the initially generated radical concentration [R]_0_ = [e_hyd_^−^]_0_.

	Total ^1^	(G)^+● 2^	(G-H2)^● 2^	(G-H1)^● 2^	Error
3 µs	1.00	0.60	0.40	0	±0.05
50 µs	0.80	0.40	0.35	0.05 ^3^	±0.1
1.4 ms	0.50	0.0	0.0	0.5	±0.1
10 ms	0.20	0.0	0.0	0.2	±0.1

^1^ from the decays at 512 nm (Figure 8); ^2^ from reconstruction of the time-resolved spectra (Figure 6); ^3^ from the rise in Figure 8.

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
