# Peer review of "Guanine Radicals Generated in Telomeric G-Quadruplexes by Direct Absorption of Low-Energy UV Photons: Effect of Potassium Ions"

_molecules, 2020, doi:10.3390/molecules25092094_

Round 1

Reviewer 1 Report

It is known that G-quadruplexes are four-stranded DNA structures that are important in many biological functions and  in the field of nanoelectronics. Thus, it is of fundamental importance to characterize the generation and fate of Guanine radicals within these systems. In this work, using nanosecond transient absorption spectroscopy, the authors carefully examined the ejected electrons and guanine radicals generated in the GGG(TTAGGG)3/K+ G-quadruplexes, following photo-ionization by low-energy 266 nm UV radiation. By comparing the results with that of several G-quadruplexes in the presence of Na+ or K+, reported by their group previously, the effect of replacing Na+ by K+ on the quantum yield of telomeric G-quadruplexes to photo-eject an electron and the decay of the overall population of guanine radicals, was further revealed. These results provide interesting mechanistic insights to understand the UV-induced photoreaction of G-quadruplex structures.

As is mentioned in the manuscript: “Unlike the tetramolecular G-quadruplexes studied by the authors’ group previously, whose structure does not change when Na+ ions are replaced by K+ ions, those formed by the human telomeric sequences undergo drastic geometrical rearrangements. A mixture of hybrid forms, depending on the ending groups, may thus be present in solution”, thus it is easy to think that in addition to the effect of K+ ions, the mixture of hybrid forms also play a role in affecting the quantum yield of telomeric G-quadruplexes to photo-eject an electron and the decay of the overall population of guanine radicals. Although the authors have addressed in this manuscript “…the exact structures adopted by the latter system is not a particular issue”, I think it would be better if the author could give more discussions about this point.

Author Response

It is known that G-quadruplexes are four-stranded DNA structures that are important in many biological functions and  in the field of nanoelectronics. Thus, it is of fundamental importance to characterize the generation and fate of Guanine radicals within these systems. In this work, using nanosecond transient absorption spectroscopy, the authors carefully examined the ejected electrons and guanine radicals generated in the GGG(TTAGGG)3/K+ G-quadruplexes, following photo-ionization by low-energy 266 nm UV radiation. By comparing the results with that of several G-quadruplexes in the presence of Na+ or K+, reported by their group previously, the effect of replacing Na+ by K+ on the quantum yield of telomeric G-quadruplexes to photo-eject an electron and the decay of the overall population of guanine radicals, was further revealed. These results provide interesting mechanistic insights to understand the UV-induced photoreaction of G-quadruplex structures.

As is mentioned in the manuscript: “Unlike the tetramolecular G-quadruplexes studied by the authors’ group previously, whose structure does not change when Na+ ions are replaced by K+ ions, those formed by the human telomeric sequences undergo drastic geometrical rearrangements. A mixture of hybrid forms, depending on the ending groups, may thus be present in solution”, thus it is easy to think that in addition to the effect of K+ ions, the mixture of hybrid forms also play a role in affecting the quantum yield of telomeric G-quadruplexes to photo-eject an electron and the decay of the overall population of guanine radicals. Although the authors have addressed in this manuscript “…the exact structures adopted by the latter system is not a particular issue”, I think it would be better if the author could give more discussions about this point.

We added the following part in the revised version including two references:

NMR experiments on G-quadruplexes formed by sequences containing four telomer repeat TTAGGG in the presence of K+ ions evidenced the coexistence of two hybrid forms in equilibrium [17, 18]. The two forms differ in the successive order of the loop arrangement and the strand orientation. The favored hybrid form depends on the capping groups.

Reviewer 2 Report

This manuscript addresses the ion dependent guanine radicals behavior in G-quadruplexes. It is based on results previously reported in the literature by the authors using quadruplexes of different sequences. Here, the same G-quadruplex sequence was considered using K+ or Na+ as cation.

This is a well-written paper, the state of the art and the aims are clearly exposed and the obtained results fit well with the conclusions. This work can be accepted after taking into account the following minor points:

-In the summary, “decades” should be changed by “order” in the sentence “ radical cations extends over four decades”

- References should be added for the values given in Table 1

- In the experimental part, a concentration should be provided for the G-quadruplex stock solution, together with a reference that establishes that the G-quadruplex structure was obtained under these conditions.

Author Response

This manuscript addresses the ion dependent guanine radicals behavior in G-quadruplexes. It is based on results previously reported in the literature by the authors using quadruplexes of different sequences. Here, the same G-quadruplex sequence was considered using K+ or Na+ as cation.

This is a well-written paper, the state of the art and the aims are clearly exposed and the obtained results fit well with the conclusions. This work can be accepted after taking into account the following minor points:

-In the summary, “decades” should be changed by “order” in the sentence “ radical cations extends over four decades”

- References should be added for the values given in Table 1

- In the experimental part, a concentration should be provided for the G-quadruplex stock solution, together with a reference that establishes that the G-quadruplex structure was obtained under these conditions.

Our answer

  • We replaced “decade” by “order of magnitude”
  • We added references in Table 1
  • We added in the revised manuscript:

“oligomer concentration: ~10-3 molL-1
and also
Prior to time-resolved experiments, TEL21/K+ solutions, kept in -20°C, were heated to 40°C, to destroy possible higher order aggregates [29], and cooled slowly to room temperature and their absorbance was adjusted 2.5 ± 0.2 over 1 cm. A typical melting curve is shown in SI-4. temperature and their absorbance was adjusted 2.5 ± 0.2 over 1 cm. A typical melting curve is shown in SI-4.